# Sensitive circulating tumor DNA–based residual disease detection in epithelial ovarian cancer

Heini ML Kallio[1],*, Kalle Savolainen[2],*, Tuomo Virtanen[1],*, Lauri Ryyppö[1],*, Hanna Selin[1], Päivi Martikainen[1], Synnöve Staff[2], Kati Kivinummi[1], Joonatan Sipola[1], Juuso Vuorinen[1], Jussi Nikkola[3], Matti Nykter[1], Annika Auranen[2], Matti Annala[1]

Epithelial ovarian cancer (EOC) is one of the leading causes of cancer-related death in women worldwide, and is characterized by a high rate of recurrence after surgery and chemotherapy. We sought to implement a circulating tumor DNA (ctDNA)–based blood test for more accurate post-operative surveillance of this disease. We analyzed 264 plasma samples collected between June 2016 and September 2021 from 63 EOC patients using tumor-guided plasma cell-free DNA analysis to detect residual disease after treatment. Assay specificity was verified using cross-patient analysis of 1,195 control samples. ctDNA was detected in 51 of 55 (93%) samples at diagnosis, and 18 of 18 (100%) samples at progression. Positive ctDNA in the last on-treatment sample was associated with rapid progression (median 1.02 versus 3.38 yr, HR = 5.63, $P < 0.001$) and reduced overall survival (median 2.31 versus NR yr, HR = 8.22, $P < 0.001$) in patients with high-grade serous cancer. In the case of 12 patients, ctDNA assays detected progression earlier than standard surveillance, with a median lead time of 5.9 mo. To approach the physical limits of ctDNA detection, five patients were analyzed using ultra-sensitive assays interrogating 479–1,856 tumor mutations, capable of tracking ctDNA fractions down to 0.0004%. Our results demonstrate that ctDNA assays achieve high sensitivity and specificity in detecting post-operative residual disease in EOC.

## Introduction

Epithelial ovarian cancer (EOC) and its most common subtype, high-grade serous cancer (HGSC), are among the most lethal cancers in women. Standard treatment for newly diagnosed EOC is surgical cytoreduction and platinum–taxane combination chemotherapy with or without angiogenesis inhibitor bevacizumab, followed by maintenance therapy using PARP inhibitors and/or bevacizumab (1). The recent adoption of PARP inhibitors as maintenance therapy has resulted in significant improvements in progression-free and overall survival, particularly for patients with homologous recombination–deficient disease (2, 3, 4). EOC is generally responsive to first-line therapy, but more than 50% of patients progress within 2 yr, even among homologous recombination–deficient patients treated with PARP inhibitors (2, 3).

Cancer antigen 125 (CA125) protein is the most used tumor marker for EOC, used in conjunction with computed tomography (CT) and transvaginal ultrasound in monitoring treatment responses and detecting tumor relapse (5). However, CA125 is not specific to ovarian cancer, as it is secreted by several organs, is elevated in 1% of cancer-free women, and is affected by menopausal status and conditions such as endometriosis and coronary artery disease (6, 7, 8). CA125 also suffers from poor sensitivity, as 20–30% of newly diagnosed EOC patients do not show elevated levels of the protein (9, 10). Treatment response monitoring is complicated by the fact that CA125 has a half-life of 15 d in the bloodstream (11). A large randomized clinical trial published in 2010 found that early initiation of chemotherapy at the first indication of CA125 progression did not improve overall survival relative to treatment initiation at symptomatic or clinical relapse (12). Because of these shortcomings, the use of CA125 in EOC surveillance is considered optional according to National Comprehensive Cancer Network guidelines (1).

Plasma circulating tumor DNA (ctDNA) is an emerging minimally invasive analyte with applications in cancer diagnosis, genomic characterization, and residual disease detection (13, 14). A highly sensitive ctDNA test could eliminate the need for periodic CT scans and gynecological examinations in EOC surveillance. Several studies have sought to implement such a test using digital droplet PCR assays targeting a single tumor mutation, with a particular focus on *TP53* mutations that occur in 99% of HGSC tumors (15, 16, 17). Pereira et al analyzed blood samples from 44 gynecological cancer patients and found detectable ctDNA in 91% of samples coinciding with a positive CT scan (18). Parkinson et al found ctDNA

[1]Faculty of Medicine and Health Technology, Tampere University and Tays Cancer Center, Tampere, Finland    [2]Department of Obstetrics and Gynecology, Tays Cancer Centre, Tampere University Hospital, Tampere, Finland    [3]Department of Urology, Tampere University Hospital, Tampere, Finland

Correspondence: annika.auranen@pirha.fi; matti.annala@tuni.fi
*Heini ML Kallio, Kalle Savolainen, Tuomo Virtanen, and Lauri Ryyppö contributed equally to this work

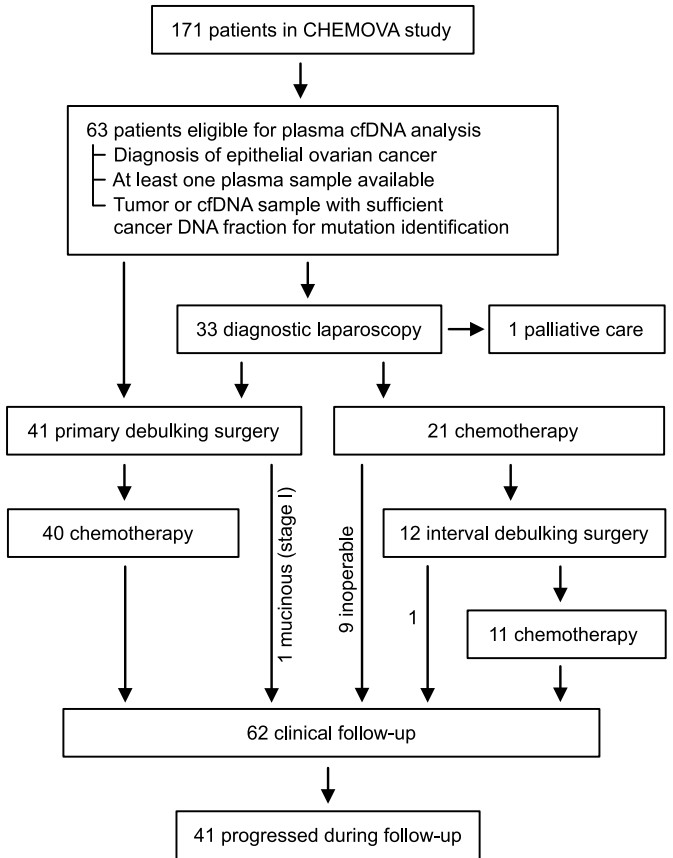

**Figure 1.  Consort diagram.**

in 42 of 51 (82%) plasma samples collected after relapse on first-line treatment ([19]). Minato et al detected ctDNA in 6 of 6 relapsed HGSC patients ([20]). Kim et al reported on-treatment ctDNA to be strongly correlated with time to progression, a finding corroborated by other studies ([18], [19], [20], [21]).

Residual disease assays targeting only a single mutation have their sensitivity fundamentally limited by the number of plasma cell-free DNA (cfDNA) fragments informing on that genomic position in a blood sample. Assays tracking multiple tumor mutations can achieve exquisite sensitivity with the aid of cross-strand DNA error correction ([22], [23]). To test the applicability of this technology for ovarian cancer surveillance, we carried out residual disease detection in 63 EOC patients using cost-effective panels targeting a median of 21 tumor mutations, as well as ultra-sensitive panels targeting up to 1,856 mutations.

# Results

### Study population

From the CHEMOVA cohort of 171 patients, 63 epithelial ovarian cancer patients were included in our retrospective analysis based on the availability of tumor tissue and plasma samples ([Fig 1]). Baseline characteristics were representative of an all-comers clinical population ([Tables 1] and S1). All patients received standard-of-care treatment consisting of primary debulking surgery (PDS) + adjuvant chemotherapy (40 patients), chemotherapy + interval debulking surgery (IDS) (12 patients), only chemotherapy (nine inoperable patients), or only primary debulking surgery (one patient with a mucinous stage I tumor). One patient received only palliative care. The median follow-up for overall survival from the start of treatment was 4.4 yr (reverse Kaplan–Meier method).

### Plasma ctDNA fraction quantification using personalized mutation assays

We collected 61 cancer tissue samples from surgery and 264 blood samples at pre- and post-operative timepoints from 63 patients (Table S2). Tumor tissue samples were sequenced to a median depth of 117x using a panel targeting 10 Mb of genomic regions, and a median of 21 tumor mutations per patient were selected for residual disease testing ([Fig 2A]). Plasma cfDNA samples were sequenced to a median depth of 2,143 fragments (IQR 1,212–3,332) per targeted mutation, with an average of 14 sequenced reads per cfDNA fragment. Plasma volume correlated with the quantity of extracted cfDNA (median 19.2 ng per ml plasma), and the amount of cfDNA used for library preparation correlated with fragment depth after sequencing (average 64 haploid genome equivalents per ng cfDNA) ([Fig 2B]). All plasma samples displayed a typical cfDNA fragment length profile with no indication of high molecular weight DNA contamination ([Fig S1]).

We used redundant cfDNA reads to produce consensus DNA sequences with a ninefold reduced error rate ([Figs 2C] and [S2]) and quantified the ctDNA fraction of each plasma sample based on the fraction of mutant DNA fragments across all target mutations. A limit of detection was established for each assay by applying it to plasma samples from other patients. Accounting for sequencing depth and background error rate, the sample-specific lowest detectable ctDNA fraction ranged between 0.003 and 0.146% (median 0.012%) ([Fig S3]). The error rate was lower for the 43% of cfDNA fragments for which both original strands were recovered, allowing cross-strand correction ([Fig 2C]). However, given the trade-off between error rate and fragment depth, we found that relaxed error correction (generating duplex consensus when possible but not requiring it) achieved the lowest limit of detection for 96% of plasma samples ([Fig S4]).

ctDNA was detected in the pre-treatment plasma of 51 of 55 (93%) patients, with ctDNA fractions between 0.018 and 41.7% (median 1.47%) (Table S3). Of the four patients with ctDNA-negative pre-treatment plasma, patients 56 and 89 had low-grade serous stage III tumors, patient 21 had a low-grade endometrioid stage II tumor, and patient 57 had a seromucinous stage I tumor. All four patients had undetectable ctDNA at all timepoints throughout the study, and no cancer recurrence during the follow-up. Pre-treatment plasma ctDNA fractions were correlated with cancer stage, type, and pre-treatment serum CA125 levels ([Fig 2D]). Plasma cfDNA samples collected after treatment initiation but before progression were available for 58 patients, and showed lower ctDNA fractions than baseline samples (median 0.001% versus 1.77%, $P = 1.8 \times 10^{-22}$, rank-sum test) ([Fig 2E and F]). To establish the specificity of our assays, we used each patient's assay to analyze samples from other patients,

**Table 1. Baseline characteristics of the 63 study patients.**

| Characteristic | Patients (%) | Median (IQR) |
|---|---|---|
| Age at diagnosis | | |
| 30–39 yr | 2 (3%) | |
| 40–49 yr | 8 (13%) | |
| 50–59 yr | 11 (17%) | |
| 60–69 yr | 19 (30%) | 67 (54–74) |
| 70–79 yr | 21 (33%) | |
| >80 yr | 2 (3%) | |
| Body mass index at diagnosis | | |
| 0–25 kg/m$^2$ | 26 (41%) | |
| 25–30 kg/m$^2$ | 19 (30%) | |
| >30 kg/m$^2$ | 17 (27%) | 26.2 (22.5–30.5) |
| Unknown | 1 (2%) | |
| CA125 at diagnosis (U/ml) | | |
| 0–35 | 2 (3%) | |
| 36–100 | 7 (11%) | |
| 101–1,000 | 34 (54%) | 509 (143–1,497) |
| >1,000 | 20 (32%) | |
| Histological classification | | |
| High-grade serous | 49 (78%) | |
| Low-grade serous | 4 (6%) | |
| Low-grade endometrioid | 5 (8%) | |
| Mucinous | 3 (5%) | |
| Carcinosarcoma | 1 (2%) | |
| Mixed cell carcinoma | 1 (2%) | |
| Stage at diagnosis | | |
| I | 5 (8%) | |
| II | 6 (10%) | |
| III | 29 (46%) | |
| IV | 23 (37%) | |
| BRCA status | | |
| BRCA1 mutant | 4 (6%) | |
| BRCA2 mutant | 4 (6%) | |
| No BRCA mutation | 47 (75%) | |
| Unknown | 8 (13%) | |
| Menopausal status | | |
| Premenopausal | 14 (22%) | |
| Postmenopausal | 49 (78%) | |

and found that 15 of 1,195 samples were falsely detected as ctDNA-positive, yielding a false-positive rate of 1.26% (Fig 2E).

Cancer cell populations that recur after treatment may not carry all of the same mutations as the biopsied tumor population because of subclone elimination. In our cohort, 13 patients had post-recurrence plasma samples with sufficient ctDNA (average mutant fragment count per conserved mutation ≥ 10) to allow us to quantify the fraction of shared mutations. We found that on average, 87% (range 63–100%) of mutations found in the biopsied population were found in post-recurrence plasma. This rate was higher for mutations that displayed a truncal allele fraction in the tissue biopsy (89% truncal versus 35% subclonal, P < 0.001, Fisher's exact test). Tumor TP53 mutations were preserved in post-recurrence plasma samples of all seven evaluable patients. In general, somatic mutations with a higher allele fraction in the tumor sample also carried a higher allele fraction in plasma (Fig S5).

## Plasma ctDNA levels decrease after surgery and chemotherapy

A total of 22 patients had plasma collected before primary debulking surgery and a second plasma sample after surgery but before starting adjuvant chemotherapy. We observed an average 84% relative decrease in ctDNA fraction after debulking surgery in these patients. The relative ctDNA% reduction and the absolute residual ctDNA% both correlated with the surgeon's assessment of the residual tumor (Fig 3). In the case of three patients (108, 122, and 155), we discovered residual ctDNA in post-PDS plasma despite surgeon-assessed lack of visible residual disease (R0), and two of these patients developed a progression after adjuvant chemotherapy. In patient 95 with metastatic disease and surgeon-assessed residual disease >1 cm (R2), we unexpectedly saw ctDNA drop to undetectable levels after surgery. The patient has not developed clinical recurrence after 44 mo of subsequent clinical follow-up on maintenance bevacizumab, has no visible disease in CT, and has maintained a CA125 level below 12 units per ml after adjuvant chemotherapy. We also carried out a similar analysis for six patients who were treated with neoadjuvant chemotherapy and IDS, but only one of these patients had detectable ctDNA after surgery (Fig S6).

## Post-operative plasma ctDNA anticipates progressions

Progression during or after first-line treatment was detected in 41 of 62 patients in the standard follow-up. Of the 43 HGSC patients who had plasma samples collected during treatment, those with detectable ctDNA in their last on-treatment sample exhibited a significantly shorter time to progression (median 1.02 versus 3.38 yr, HR = 5.63, P < 0.001) and overall survival (median 2.31 versus NR yr, HR = 8.22, P < 0.001) (Fig 4A). A similar pattern was observed when the analysis was expanded to all 55 epithelial ovarian cancer patients (Fig 4B). Surgeon-assessed residual disease after surgery was also associated with time to progression and overall survival (Fig 4C), although the prognostic association was weaker, because surgery was generally followed by adjuvant chemotherapy with variable efficacy.

Of the 21 epithelial ovarian cancer patients with detectable ctDNA in their last on-treatment sample, all but one subsequently developed a clinical recurrence (Fig 4A). The sole exception was patient 33, whose ctDNA-positive end-of-treatment plasma sample (P = 0.046, two mutant fragments) was preceded by two and followed by three ctDNA-negative samples, suggesting a false positive (Tables S2 and S3).

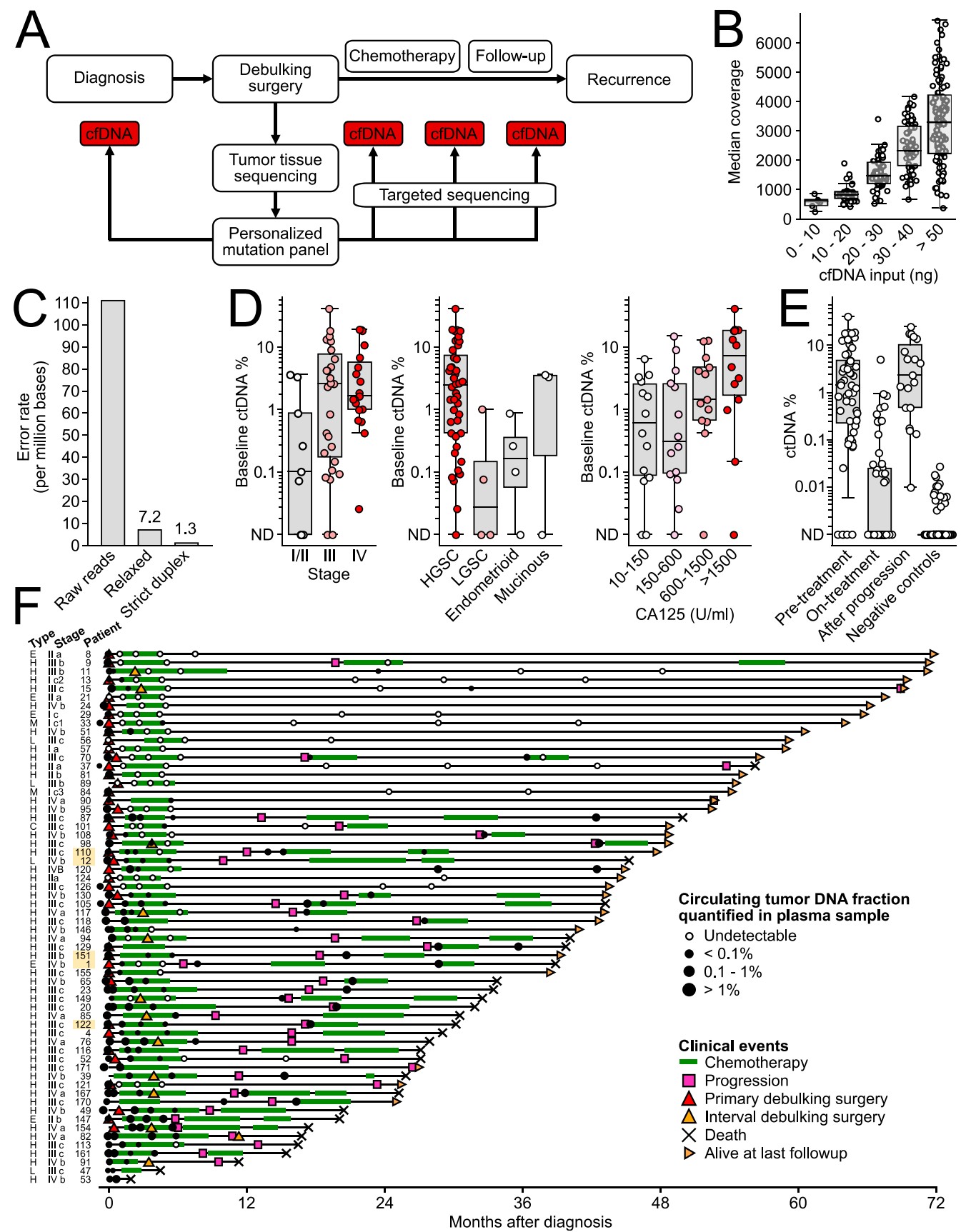

18 patients had a plasma sample collected soon after progression (9–100 d after progression, median 37 d). A positive ctDNA fraction was found in all 18 samples, with ctDNA fractions between 0.006 and 25.3% (median 2.1%). Based on the robust ctDNA levels at progression, we investigated 22 patients who had plasma collected between the end of first-line treatment (up to 1 wk before the end of chemotherapy) and progression. A positive ctDNA sample preceding the progression was found for 12 (55%) of these patients, anticipating the progression by 14–1,417 d (median 179) (Fig 4D).

### Plasma ctDNA fraction correlates with CA125 measurements

For comparison with plasma ctDNA, we also evaluated changes in serum CA125 concentration as a predictor of clinical recurrence. GCIG criteria for CA125 progression (see the Materials and Methods section) were met by only 17 of 41 (41%) patients before progression, and 2 of 41 (5%) patients before the end of first-line treatment, consistent with previous studies (24). GCIG criteria for CA125 response were met by 95% of patients during first-line treatment, 68% of whom later progressed clinically (Table S1). We also tested dividing the patients into two groups based on whether their serum CA125 level was within the normal reference range (<35 units per ml) at the end of first-line treatment. Surprisingly, this simple criterion achieved prognostic performance close to that obtained using our targeted ctDNA assays (Fig S7A and B). Overall, we observed a strong correlation between CA125 and ctDNA% in patients with serial time-matched measurements (Fig S7C). Because of endogenous secretion of CA125 from several organs, the median CA125 level at the end of treatment was 19 units per ml even in patients who did not develop a progression during subsequent follow-up.

### Tissue WGS enables ultra-sensitive residual disease testing

Because the sensitivity of our narrower ctDNA assays targeting a median of 21 mutations was primarily limited by the number of captured DNA fragments, we decided to approach the physical limits of mutation-based residual disease testing using larger panels. We performed whole-genome sequencing on tumor samples from five patients (1, 12, 110, 122, and 151) who had progressed clinically but had post-operative plasma samples where ctDNA was undetectable using the narrower assays. We selected 479–1,856 somatic mutations per patient for ultra-sensitive residual disease testing, and used hybridization capture panels to sequence a total of 14 post-operative plasma samples (two to four per patient) to a median duplex consensus fragment depth of 595x (Fig 5A, Table S4). The larger assays significantly outperformed their smaller counterparts, achieving a median limit of detection of 0.0010% ctDNA (equivalent to five mutant fragments per million). The lowest theoretical limit of detection for any sample was 0.0004% ctDNA,

and the lowest ctDNA fraction measured in any sample was 0.00086% (8/1,849,921 mutant DNA fragments). Of the 14 post-operative cfDNA samples analyzed using the ultra-sensitive assays, 11 (79%) were found to carry a detectable ctDNA fraction (Fig 5A). In three of the five patients (122, 151, and 12), the ultra-sensitive assays allowed the ctDNA fraction to be detected and quantified at all plasma timepoints. To determine the false-positive rate of the larger panels, we analyzed 18 cross-patient control samples and found zero false positives (Fig 5B). Because of the larger number of captured informative fragments, strict duplex consensus error correction was found to be superior to relaxed consensus for all samples analyzed using the larger assays (Fig S4). Interestingly, we observed that 0.7% of targeted mutations displayed an anomalously high allele fraction inconsistent with other tumor mutations. We hypothesized that these anomalies represented mosaicism, technical artifacts, amplified mutations, or clonal hematopoiesis (CHIP), and omitted them from analysis using automated outlier detection (see the Materials and Methods section).

As an interesting anecdote, patient 1 had developed a suspicious 1 cm lump in their groin on their fifth cycle of chemotherapy, but a blood sample collected at this time was ctDNA-negative (1/368,251 mutant DNA fragments). 2 mo later, the lump had increased in size and needle biopsy identified it as a lymph node metastasis. In a blood sample collected 37 d after the biopsy-confirmed progression, the ctDNA fraction had risen to 0.137%, representing a 252-fold increase in 91 d (Fig 5A). This case demonstrates that patients may carry extremely low ctDNA fractions in blood samples collected only 2 mo before clinically detectable progression.

## Discussion

In this retrospective analysis, we show that tumor-guided plasma ctDNA analysis enables post-operative residual cancer detection at sensitivities as low as two mutant DNA molecules per million (i.e., 0.0004% ctDNA fraction) in EOC patients. Cost-effective assays targeting a median of 21 tumor mutations were able to detect ctDNA in all 18 patients who had plasma collected after progression, suggesting that ctDNA assays could potentially replace contemporary EOC surveillance that relies on some combination of CA125 blood tests, CT scans, gynecological examinations, and transvaginal ultrasonography. Unlike CA125-based surveillance that suffers from background secretion from several organs and only detected 41% of progressions in this cohort, plasma ctDNA mutations represent a biomarker with near-zero background. This allows ctDNA-based testing to achieve a high level of sensitivity while maintaining a low rate of false positives. A recent study using the multiplex PCR Signatera assay also reported successful detection of progression events in 7 of 7 EOC patients (25). Plasma ctDNA-based residual

**Figure 2. Residual disease testing in epithelial ovarian cancer using a circulating tumor DNA (ctDNA)–based approach.**
**(A)** Overview of tumor tissue–guided residual disease testing. **(B)** Boxplot showing how the amount of plasma cell-free DNA used for library preparation affected median fragment coverage per mutation. **(C)** Effect of two different DNA error correction methods on background error rate. **(D)** Relationship between ctDNA fraction and cancer stage, cancer type, and CA125 level at diagnosis. **(E)** Comparison of ctDNA fractions in plasma samples collected before treatment, on-treatment, and after progression, as well as 1,195 negative control samples from other patients. **(F)** Visualization of patient clinical timelines and ctDNA fractions in serial plasma samples (circles). Progression was defined as a positive CT scan or biopsy-proven metastasis (see the Materials and Methods section). For patients 1, 12, 110, 122, and 151 indicated with a yellow background, plasma ctDNA fractions quantified using ultra-sensitive assays (see Fig 5) are shown.

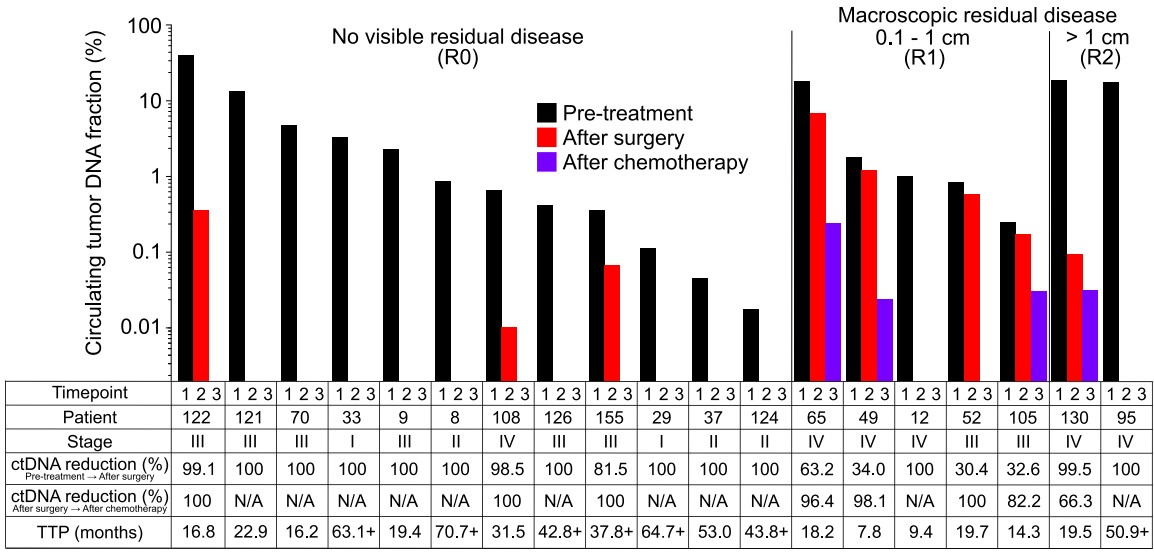

| Timepoint | 1 2 3 | 1 2 3 | 1 2 3 | 1 2 3 | 1 2 3 | 1 2 3 | 1 2 3 | 1 2 3 | 1 2 3 | 1 2 3 | 1 2 3 | 1 2 3 | 1 2 3 | 1 2 3 | 1 2 3 | 1 2 3 | 1 2 3 | 1 2 3 | 1 2 3 |
|---|---|---|---|---|---|---|---|---|---|---|---|---|---|---|---|---|---|---|---|
| Patient | 122 | 121 | 70 | 33 | 9 | 8 | 108 | 126 | 155 | 29 | 37 | 124 | 65 | 49 | 12 | 52 | 105 | 130 | 95 |
| Stage | III | III | III | I | III | II | IV | III | III | I | II | II | IV | IV | IV | III | III | IV | IV |
| ctDNA reduction (%) Pre-treatment → After surgery | 99.1 | 100 | 100 | 100 | 100 | 100 | 98.5 | 100 | 81.5 | 100 | 100 | 100 | 63.2 | 34.0 | 100 | 30.4 | 32.6 | 99.5 | 100 |
| ctDNA reduction (%) After surgery → After chemotherapy | 100 | N/A | N/A | N/A | N/A | N/A | 100 | N/A | 100 | N/A | N/A | N/A | 96.4 | 98.1 | N/A | 100 | 82.2 | 66.3 | N/A |
| TTP (months) | 16.8 | 22.9 | 16.2 | 63.1+ | 19.4 | 70.7+ | 31.5 | 42.8+ | 37.8+ | 64.7+ | 53.0 | 43.8+ | 18.2 | 7.8 | 9.4 | 19.7 | 14.3 | 19.5 | 50.9+ |

**Figure 3. Reduction in plasma circulating tumor DNA fraction after primary debulking surgery and adjuvant chemotherapy in 19 patients.**
Patients who did not have all three plasma timepoints available (n = 18) or had undetectable circulating tumor DNA at all three timepoints (n = 3) were omitted from the visualization. Right-censored time-to-progression values are indicated with a plus sign.

disease testing can be a favorable alternative to CT imaging, as it can be cheaper and does not expose patients to ionizing radiation.

Beyond small and cost-effective assays, we show that larger ultra-sensitive assays targeting up to 1,856 tumor mutations can achieve a very low limit of detection for ctDNA, tracking otherwise undetectable cancer cell populations in a patient's body after surgery and chemotherapy. The sensitivity of ctDNA assays depends on the number of tumor-informative DNA fragments and the assay's background error rate. Previous studies have demonstrated that base substitution errors can be drastically reduced via computational error correction between complementary DNA strands (22, 23), with the caveat that both strands are only recovered for a minority of plasma cfDNA fragments by current library construction protocols (22). We found that assays interrogating less than 200,000 tumor-informative DNA fragments can achieve better sensitivity using relaxed DNA error correction that does not discard non-duplex DNA fragments.

An average blood sample from a cancer patient contains millions of DNA fragments that are informative of tumor mutation allele fractions, but detecting cancer fractions lower than one mutant fragment per million requires extremely low background error rates. In our cohort, we measured average background base error rates of 0.0007% and 0.0001% for relaxed consensus and strict duplex consensus fragments, respectively. Phased mutations have lower background error rates, but are less abundant in tumor genomes (26). Novel library construction protocols that avoid complementary strand synthesis during DNA repair can reduce duplex error rates by preventing single-stranded DNA damage from being propagated into the complementary strand (27, 28).

Detectable ctDNA at the end of first-line treatment was strongly associated with shorter time to progression and overall survival in our cohort, confirming previous findings (18, 19, 20, 21). This implies that residual ctDNA status could potentially be used for optimizing treatment intensity. In 2010, Rustin et al showed that early initiation of second-line chemotherapy based on serum CA125 monitoring offered no survival benefit in a 1:1 randomized trial of 1442 EOC patients (12). Future studies may re-evaluate this hypothesis with new therapeutic agents and the improved accuracy afforded by ctDNA assays. In conclusion, we have demonstrated that ctDNA detection from a few milliliters of blood offers an objective, highly sensitive, and minimally invasive method for residual disease detection in EOC.

## Materials and Methods

### Patient cohort

Patients with an ovarian mass diagnosed and treated at the Department of Obstetrics and Gynecology, Tampere University Hospital, Finland, were recruited to the CHEMOVA study cohort between May 2016 and February 2020 (ClinicalTrials.gov: NCT02758652). All patients provided written informed consent before participating in the study. The study was conducted in accordance with the Declaration of Helsinki, and the protocol was approved by the Regional Ethics Committee of Tampere University Hospital (R15134, 1 September 2015).

Treatment was based on standard clinical practice. Patients underwent either PDS followed by adjuvant chemotherapy (six cycles of paclitaxel + carboplatin) (29), or neoadjuvant chemotherapy (three to four cycles of paclitaxel + carboplatin) followed by IDS and adjuvant chemotherapy (up to six to eight cycles). Operability was evaluated based on the patient condition, imaging, and Fagotti score (30). Inoperable patients received only chemotherapy. EOC diagnosis was confirmed by morphological and immunohistochemical analyses of tumor samples. In FIGO stage IV disease or stage III disease with suboptimal surgery, up to 22 cycles of bevacizumab were combined with paclitaxel–carboplatin chemotherapy (31, 32). Serum CA125 levels were recorded at diagnosis and

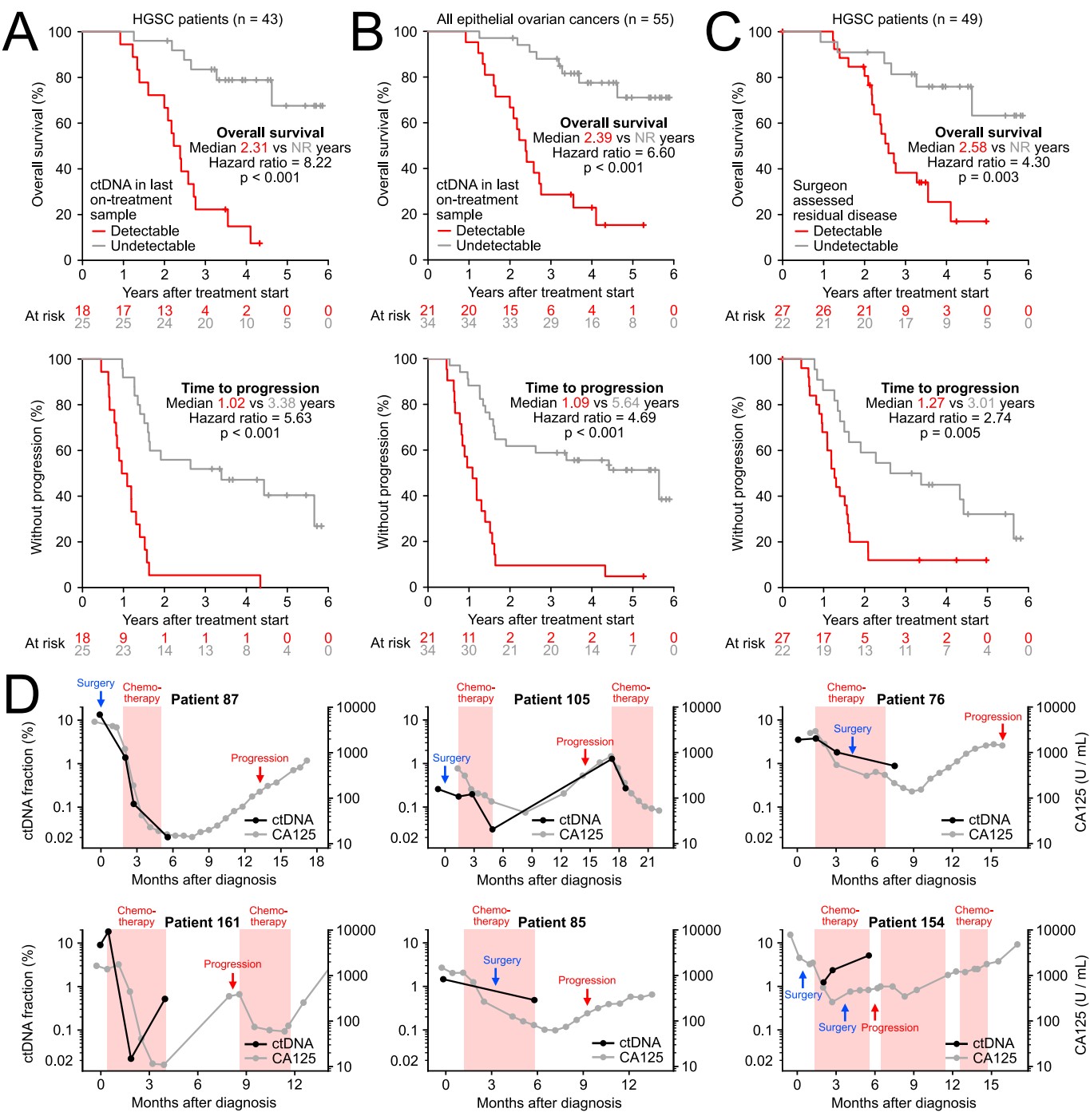

**Figure 4. Detectable circulating tumor DNA (ctDNA) at the end of first-line treatment associates with overall survival and time to progression in epithelial ovarian cancer patients.**

**(A, B, C)** Kaplan–Meier analysis comparing (A) high-grade serous ovarian cancer patients with and without detectable ctDNA at the end of first-line treatment, (B) all epithelial ovarian cancer patients with and without detectable ctDNA at the end of first-line treatment, and (C) high-grade serous cancer patients with and without surgeon-assessed residual disease after debulking surgery. **(D)** Six representative patient cases where detectable plasma ctDNA at the end of first-line treatment indicated residual disease and anticipated a subsequent progression. Black dots indicate ctDNA fraction measurements. Gray dots indicate CA125 measurements. Linear interpolation lines are shown for both. All plasma ctDNA fractions shown in this figure were quantified using the narrow residual disease assays targeting a median of 21 tumor mutations.

throughout treatment according to patient preference. A body CT scan was obtained at diagnosis, and also after chemotherapy for patients with residual or inoperable tumors.

Standard post-treatment follow-up included clinical examination with gynecological sonography four times in year 1, three times in year 2, and two times in year 3. If disease

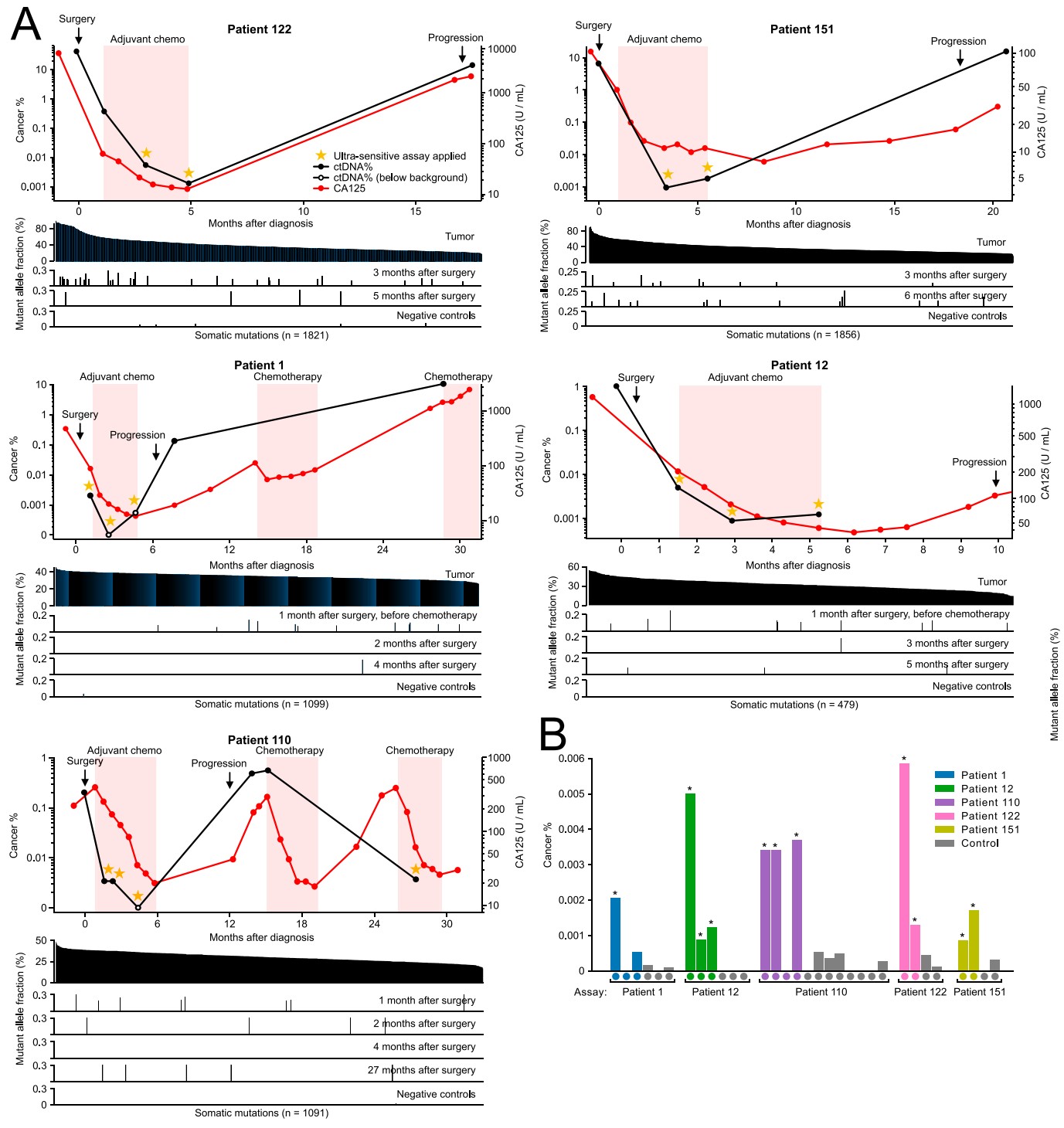

**Figure 5. Post-operative detection of residual circulating tumor DNA using large mutation panels designed based on whole-genome sequencing of tumor tissue.**
**(A)** Circulating tumor DNA fractions (black) and CA125 levels (red) in serial blood draws. Dots indicate measurements. Bar plots underneath indicate allele fractions of individual tumor mutations in the plasma cell-free DNA samples. Most bars represent a single mutant DNA fragment. **(B)** Cancer fractions quantified in patient plasma samples (colored bars) and control samples from other patients (gray). Samples are grouped by assay. A star symbol above a bar indicates that the cancer fraction was statistically significantly higher ($P < 0.05$) than the background error level.

progression was suspected, a body CT scan was obtained. Progression after initiation of first-line treatment was defined according to RECIST 1.1 criteria as a 20% increase in total lesion

size or a new lesion visible in the CT scan, or a biopsy-proven new metastasis. Elevation of CA125 alone was not counted as progression.

### Analysis of clinical outcomes

Time to progression was defined as the number of days from first-line treatment initiation to the date of progression. Overall survival was defined as the number of days from first-line treatment initiation to the last day of follow-up. CA125 progression was defined according to Gynecological Cancer Intergroup (GCIG) criteria as a CA125 level greater or equal to two times the nadir or upper limit of the CA125 reference range (whichever is higher), measured on two occasions at least 1 wk apart (33).

### Plasma cfDNA collection

Plasma for ctDNA measurement was scheduled to be collected at diagnosis (pre-surgery), at first, third, and sixth cycles of chemotherapy, at progression, and during follow-up at 1-yr intervals after completion of first-line treatment. At each timepoint, 10 ml of blood was drawn into an EDTA tube. The plasma and cell fraction were separated by centrifugation within 2 h of sample collection and stored at –70°C. cfDNA was extracted from 0.8–5.0 ml of plasma (median 2.0 ml) using QIAGEN Circulating Nucleic Acid kits and quantitated using Qubit dsDNA high-sensitivity assays (Thermo Fisher Scientific).

### Tissue DNA and RNA extraction

Tumor samples were collected during PDS, IDS, or diagnostic procedures into tubes containing Tissue-Tek compound (Sakura Finetek), snap-frozen in liquid nitrogen, and stored at –70°C. DNA was extracted using the QIAGEN AllPrep DNA/RNA/miRNA Universal kit or QIAamp FAST Tissue DNA kit, and quantitated using Qubit dsDNA broad range assays (Thermo Fisher Scientific). For eight tissue samples, DNA and tissue had been used up for a different study, but RNA remained available and was used for mutation detection.

### Residual disease testing

Sequencing libraries of tumor tissue samples were subjected to hybridization capture using the IDT xGen Inherited Disease panel, which captures 10 Mb of genomic regions focused on coding regions and cancer-associated genes. The tissue samples were analyzed for the presence of single nucleotide and short insertion/deletion variants not found in the gnomAD 3.0 human SNP database, and having a low background error rate in a set of 17 cancer-negative control plasma cfDNA samples. Custom IDT xGen Discovery hybridization capture panels targeting 15–166 candidate mutations per patient were ordered. To enable parallel processing of multiple cfDNA samples in each hybridization capture reaction, each individual panel was designed to target mutations from two to seven patients. This approach allowed us to generate a large number of negative control samples, as each patient's residual disease assay was used to analyze plasma samples from several patients. All libraries were sequenced using Illumina NovaSeq 6000 instruments, using v1.5 paired-end 2 × 150 cycle kits. The computational analysis of sequenced libraries, including alignment, DNA error correction, and residual disease detection, is described in Supplementary Methods.

### Statistical analysis

Cox regression analyses and Kaplan–Meier curves were calculated using R (version 4.2.2) with the "survival" package (version 3.4.0). All other statistical tests and data analyses were conducted in Julia v1.8.2 with packages HypothesisTests 0.10.11 and Distributions 0.25.76.

## Data Availability

Raw sequencing data have been deposited into the European Genome–Phenome Archive (EGA) under study accession EGAD50000000360. All somatic mutations used for ctDNA quantification, as well as fragment counts and background error rates, are provided as Supplementary Tables.

## Supplementary Information

## Acknowledgements

This work was funded by the Jane and Aatos Erkko Foundation, Academy of Finland Center of Excellence Programme (project 312043), Competitive State Research Financing of the Expert Responsibility Area of Tampere University Hospital, Finnish Cultural Foundation, and TAYS Foundation. The authors acknowledge the Biocenter Finland (BF) and Tampere Genomics Facility for their service. No funding sources were involved in the design or execution of the study. We are grateful to all participating patients and their families.

### Author Contributions

HML Kallio: conceptualization, investigation, visualization, methodology, and writing—review and editing.
K Savolainen: conceptualization, data curation, funding acquisition, investigation, methodology, and writing—original draft.
T Virtanen: data curation, software, formal analysis, validation, investigation, visualization, methodology, and writing—review and editing.
L Ryyppö: data curation, validation, investigation, visualization, and writing—review and editing.
H Selin: data curation, investigation, and methodology.
P Martikainen: data curation, investigation, and methodology.
S Staff: conceptualization, data curation, funding acquisition, investigation, and writing—review and editing.
K Kivinummi: data curation and investigation.
J Sipola: data curation, software, investigation, and visualization.
J Vuorinen: investigation and methodology.
J Nikkola: data curation, investigation, and writing—review and editing.

M Nykter: resources, funding acquisition, project administration, and writing—review and editing.

A Auranen: conceptualization, data curation, supervision, funding acquisition, investigation, project administration, and writing—review and editing.

M Annala: conceptualization, data curation, software, formal analysis, supervision, funding acquisition, validation, investigation, visualization, methodology, and writing—original draft, review, and editing.

## Conflict of Interest Statement

HML Kallio, J Vuorinen, and M Annala report shareholder positions in Fluivia. No disclosures were reported by the other authors.

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
