## [Reviewer comments · Life Science Alliance]

Life Science Alliance

Sensitive circulating tumor DNA based residual disease detection in epithelial ovarian cancer

Heini Kallio, Kalle Savolainen, Tuomo Virtanen, Lauri Ryyppö, Hanna Selin, Päivi Martikainen, Synnöve Staff, Kati Kivinummi, Joonatan Sipola, Juuso Vuorinen, Jussi Nikkola, Matti Nykter, Annika Auranen, and Matti Annala

DOI: <https://doi.org/10.26508/lsa.202402658>

Corresponding author(s): Matti Annala, Tampere University and Annika Auranen, Tampere University Hospital

Review Timeline:

Submission Date:	2024-02-14
Editorial Decision:	2024-02-14
Revision Received:	2024-02-23
Editorial Decision:	2024-02-27
Revision Received:	2024-03-15
Accepted:	2024-03-18

Transaction Report:

Please note that the manuscript was previously reviewed at another journal and the reports were taken into account in the decision-making process at *Life Science Alliance*.

Reviews

Referee #1

Remarks for Author:

In the current manuscript, Kallio and colleagues explore the utility of ctDNA-based disease detection for ovarian cancer. The authors apply what sounds like an in-hour tumor-informed ctDNA-MRD approach. The authors use an off-the-shelf 10Mb IDT panel to identify tumor-specific mutations, and then sequence these mutations to ultra-high unique molecular depth. The authors demonstrate that ctDNA has a number of areas of potential clinical utility, including correlation with stage, cancer histology, CA-125, and residual disease status as estimated by surgeon. The authors also show that residual ctDNA detection at the completion of therapy is prognostic for clinical outcomes, including overall survival. Finally, the authors demonstrate that utilizing a ctDNA assay that tracks more mutations, as identified by WGS, leads to. The manuscript is well written and addresses an important topic in oncology. The use of tumor-informed ctDNA has been demonstrated in many prior studies for other cancer types, including colorectal cancer, lung cancer, breast cancer, and lymphomas. The concept that tracking more mutations increases sensitivity is also not in and of itself novel, as this is well established in the field. However, the specific application to ovarian cancer is less well-established, which does afford some novelty to this study.-

Overall I find the results of the manuscript well thought-out and well supported. I do have questions about the technical details of this manuscript which should be answered to improve the clarity and interpretability of the manuscript.

1. My major question about this regarding the use of UMIs. In the "Sequencing Library Construction" section, there is mention of dual sample index barcodes, but there is no mention of UMIs. The UMIs are only mentioned when the authors get to the consensus-based DNA fragment error correction. Were UMIs used? If so, which UMIs were used?
2. Again regarding the use of UMIs, the authors state that they used duplex consensus reads in a "relaxed" fashion, similar to what was done in the iDES-enhanced CAPP-Seq method (Newman et al, Nature Biotechnology 2016). More information regarding the UMI scheme is needed to understand what level of error correction is truly possible here.
3. I see that the authors did not include germline sequencing. It is now well established that paired-germline is required to control for clonal hematopoiesis for ctDNA sequencing (Razavi et al, Nature Medicine 2019). How are the authors sure that the variants they are tracking tumor mutations as opposed to CH?

Referee #2

Comments on Novelty/Model System for Author:

Although ctDNA is an area that is currently attracting much interest and attention from readers, there have already been many previous reports. The content of the authors' report is basically the same as that of the previous report, and in terms of novelty, it is insufficient for publication in a high-impact journal such as this one.

Remarks for Author:

This study explores the clinical significance of ctDNA. It shows that the sensitivity can be greatly improved by using the genetic mutation information of the tumor, and that ctDNA can be detected in cases that were difficult to identify with conventional techniques.

Although ctDNA is a field that is currently attracting a great deal of attention, there have already been many previous reports. The content of the authors' report is basically the same as that of the previous report, and in terms of novelty, it is insufficient for publication in a high-impact journal such as this.

The attempt to detect a small fraction of ctDNA using a composite panel of tumor mutations is interesting, but there are already several prior reports on this topic and it is not novel. Therefore, it cannot be said that this study adds new information to the field of ctDNA analysis.

Transferred Consultation Comments

- Thank you for your comments. As you say, it is not yet common in ovarian cancer, and there may be a potentials of interest about the UMIs used. As for novelty, I would like to leave the final decision to the editor. Certainly, I get the impression that the clinical samples have been carefully investigated.
- Overall I thought this manuscript was only of medium impact. Certainly, as you say, there are quite a number of studies on ctDNA as a prognostic tool, although there is less of this in Ovarian cancer than in other cancers. So while the concept of tracking ctDNA is not novel, the applications in Ovarian cancer are of some interest. The technical aspects need to be significantly fleshed out in my mind, as there is essentially no detail about the UMIs used, but from a novelty standpoint I thought the application in ovarian cancer added at least some novelty.
- Thank you for choosing me as a reviewer. I read your other comments with interest.

There are several papers on ctDNA and most of them state that ctDNA is a sensitive predictor of recurrence. I think this study did a lot of research on how to identify ctDNA, but I felt that the final conclusion was that it is a prognostic tool, which is not very new from a clinical point of view. If it leads to some new stratification, it would have more clinical value, though. So, overall, my opinion is that it is not novel. However, I am not familiar enough with ctDNA myself, so I may not understand the true value of the novelty they are reporting. I agree that they are doing a sufficient amount of investigation.

February 14, 2024

Re: Life Science Alliance manuscript #LSA-2024-02658-T

Mr. Matti Annala
Tampere University
Faculty of Medicine and Health Technology
Arvo Ylpön katu 34
Tampere, Pirkanmaa 33520
FINLAND

Dear Dr. Annala,

Thank you for submitting your manuscript entitled "Sensitive circulating tumor DNA based residual disease detection in epithelial ovarian cancer" to Life Science Alliance. We invite you to submit a revised manuscript addressing Reviewer 1's comments.

Thank you for this interesting contribution to Life Science Alliance. We are looking forward to receiving your revised manuscript.

Sincerely,

B. MANUSCRIPT ORGANIZATION AND FORMATTING:

Referee #1 (Remarks for Author):

In the current manuscript, Kallio and colleagues explore the utility of ctDNA-based disease detection for ovarian cancer. The authors apply what sounds like an in-hour tumor-informed ctDNA-MRD approach. The authors use an off-the-shelf 10Mb IDT panel to identify tumor-specific mutations, and then sequence these mutations to ultra-high unique molecular depth. The authors demonstrate that ctDNA has a number of areas of potential clinical utility, including correlation with stage, cancer histology, CA-125, and residual disease status as estimated by surgeon. The authors also show that residual ctDNA detection at the completion of therapy is prognostic for clinical outcomes, including overall survival. Finally, the authors demonstrate that utilizing a ctDNA assay that tracks more mutations, as identified by WGS, leads to

The manuscript is well written and addresses an important topic in oncology. The use of tumor-informed ctDNA has been demonstrated in many prior studies for other cancer types, including colorectal cancer, lung cancer, breast cancer, and lymphomas. The concept that tracking more mutations increases sensitivity is also not in and of itself novel, as this is well established in the field. However, the specific application to ovarian cancer is less well-established, which does afford some novelty to this study.-

Overall I find the results of the manuscript well thought-out and well supported. I do have questions about the technical details of this manuscript which should be answered to improve the clarity and interpretability of the manuscript.

We would like to thank the reviewer for their thorough and thoughtful review of our manuscript.

1. My major question about this regarding the use of UMIs. In the "Sequencing Library Construction" section, there is mention of dual sample index barcodes, but there is no mention of UMIs. The UMIs are only mentioned when the authors get to the consensus-based DNA fragment error correction. Were UMIs used? If so, which UMIs were used?

We used the Agilent SureSelect XT HS2 DNA library construction kit (this is mentioned in Supplementary Methods), which utilizes adapters that incorporate a 3 bp double-stranded (aka duplex) UMI immediately next to the ligation site, for a total of 3 + 3 = 6 degenerate duplex UMI bases per cfDNA fragment. A small subset of older samples (19 out of 264 samples) instead underwent library construction using the KAPA HyperPrep kit, which does not include UMIs.

The following figure from the Agilent SureSelect XT HS2 product manual (see page 47 of <https://www.agilent.com/cs/library/usermanuals/public/G9983-90000.pdf>) illustrates the structure of the duplex UMI adapters:

MBC-tagged libraries

We have now added the following sentence to the Supplementary Methods section

“Sequencing library construction” to clarify this:

“The Agilent SureSelect XT HS2 DNA kit was used with Agilent’s MBC adapters which incorporate a 3 bp duplex unique molecular identifier (UMI), for a total duplex UMI length of 6 bp per cfDNA fragment. The KAPA HyperPrep kit was used without UMIs.”

We have also added a column “Library prep kit” to Supplementary Table 2 to make it clear which samples were analyzed using the KAPA HyperPrep kit. Library construction could not be redone for the 19 KAPA HyperPrep samples using the Agilent kit because we did not have sufficient cfDNA left for a second library construction.

We also corrected the name of the library preparation kit used for the 19 samples from “Roche SeqCap EZ HyperCap” to “KAPA HyperPrep” in the Supplementary Methods. The former name was incorrect as it was actually the name of Roche’s hybridization capture kit.

2. Again regarding the use of UMIs, the authors state that they used duplex consensus reads in a “relaxed” fashion, similar to what was done in the iDES-enhanced CAPP-Seq method (Newman et al, Nature Biotechnology 2016). More information regarding the UMI scheme is needed to understand what level of error correction is truly possible here.

We hope our answer to the previous question clarifies our use of UMI adapters in library construction. Regarding the 19 KAPA HyperPrep samples analyzed without UMIs, it is critical to note that both single-strand and duplex consensus error correction can be performed in the absence of UMIs. Redundant duplicates for consensus generation are identified based on DNA fragment boundaries, and the original physical DNA strand from which each paired end sequencing read originates is inferred from the strand of the first mate. The benefit that UMIs provide is that if two original double-stranded cfDNA fragments in a sample happen to have the same boundaries (i.e. both fragments start and end at the exact same chromosomal coordinates), the UMI allows distinguishing between the fragments. This benefit is only realized when library complexity is sufficiently high that same-boundary cfDNA fragments become commonplace.

If consensus error correction is applied without UMIs and there are plasma cfDNA fragments sharing the same boundaries, allele fractions of somatic mutations with a true allele fraction < 50% will be biased down due to consensus being applied across mutant and wildtype cfDNA fragments sharing the same boundaries. In residual disease testing, this would manifest as reduced sensitivity.

At the sequencing depth we used, only a minority of cfDNA fragments shared the same boundaries. The Figure below shows the median depth at targeted mutations when consensus was applied utilizing UMIs (x-axis) and without utilizing UMIs (y-axis). Each dot is a sample that underwent library construction using the Agilent XT HS V2 kit. Depending on the sequencing depth of the sample, between 0 - 40% (and typically around 20%) of depth was lost when the sample was analyzed without UMIs. This indicates that only a minority of cfDNA fragments in each sample shared exact boundaries with another cfDNA fragment.

Consistent with this finding, we did not observe any obvious loss of sensitivity in the 19 samples analyzed without UMIs.

3. I see that the authors did not include germline sequencing. It is now well established that paired-germline is required to control for clonal hematopoiesis for ctDNA sequencing (Razavi et al, Nature Medicine 2019). How are the authors sure that the variants they are tracking tumor mutations as opposed to CH?

We agree that clonal hematopoiesis is a significant confounder when using plasma cfDNA analysis to characterize a cancer's somatic mutation profile, and we always include leukocyte DNA analysis in such studies (see e.g. Herberts & Annala et al. 2022 or Annala et al. 2021).

However, in the context of tumor-informed plasma ctDNA residual disease detection, we are not searching for *de novo* mutations in plasma cfDNA, but are instead simply quantifying the ctDNA evidence for somatic mutations found in primary tumor tissue. While leukocyte-derived cfDNA constitutes a large fraction of plasma cfDNA, leukocytes have a much smaller representation in tumor tissue. This is why plasma ctDNA residual disease detection studies (including commercial assays) do not generally analyze leukocyte DNA. That being said, there is at least one article (Severson et al. 2018) suggesting that expanded leukocyte populations may under some circumstances be sufficiently abundant in tumor tissue to result in CHIP mutations being detected in tissue samples.

In the Results section of our manuscript we originally noted that we observed some rare mutations in our cohort that behaved as outliers inconsistent with other tumor mutations, and we proposed CHIP as one possible explanation:

“Interestingly, we observed that 0.7% of targeted mutations displayed an anomalously high allele fraction (>1%) inconsistent with other tumor mutations. We hypothesized that these anomalies represented clonal hematopoiesis (CHIP), mosaicism, or amplified mutations, and omitted them from analysis using automated outlier detection (see Methods).”

In retrospect, we view CHIP as an unlikely explanation as none of these outlier mutations involved the classic CHIP mutated genes *TET2*, *DNMT3A*, *ASXL1*, or *SF3B1*. We have now added the outlier mutations to Supplementary Table 4 (indicated with the label “Rejected as outlier”) so they can be inspected by readers. We have also revised the text relating to the outlier mutations, listing CHIP later in the list of hypotheses:

“Interestingly, we observed that 0.7% of targeted mutations displayed an anomalously high allele fraction inconsistent with other tumor mutations. We hypothesized that these anomalies represented mosaicism, technical artifacts, amplified mutations, or clonal hematopoiesis (CHIP), and omitted them from analysis using automated outlier detection (see Methods).”

More broadly, the overall lack of mutations in typical CHIP genes in our cohort speaks against CHIP playing any confounding role in our study. Our Supplementary Table 3 lists one DNMT3A missense mutation and one ASXL1 intronic mutation, and both have allele fractions that follow other tumor mutations at all timepoints. Our Supplementary Table 4 does not list any mutations in CHIP genes.

February 27, 2024

RE: Life Science Alliance Manuscript #LSA-2024-02658-TR

Mr. Matti Annala
Tampere University
Faculty of Medicine and Health Technology
Arvo Ylpön katu 34
Tampere, Pirkanmaa 33520
Finland

Dear Dr. Annala,

Thank you for submitting your revised manuscript entitled "Sensitive circulating tumor DNA based residual disease detection in epithelial ovarian cancer". We would be happy to publish your paper in Life Science Alliance pending final revisions necessary to meet our formatting guidelines.

- please be sure that the authorship listing and order is correct
- please upload all figure files as individual ones, including the supplementary figure files; all figure legends should only appear in the main manuscript file after the references section
- please add ORCID ID for the secondary corresponding author -- they should have received instructions on how to do so
- please add the Twitter handle of your host institute/organization as well as your own or/and one of the authors in our system
- please add an Author Contributions section to your main manuscript text
- please add callouts for Figures 2F, S7A-C and Table S4 to your main manuscript text
- please include accession information for the raw sequencing data in the Data Availability statement

A. FINAL FILES:

B. MANUSCRIPT ORGANIZATION AND FORMATTING:

Sincerely,

March 18, 2024

RE: Life Science Alliance Manuscript #LSA-2024-02658-TRR

Mr. Matti Annala
Tampere University
Faculty of Medicine and Health Technology
Arvo Ylpön katu 34
Tampere, Pirkanmaa 33520
Finland

Dear Dr. Annala,

Thank you for submitting your Research Article entitled "Sensitive circulating tumor DNA based residual disease detection in epithelial ovarian cancer". It is a pleasure to let you know that your manuscript is now accepted for publication in Life Science Alliance. Congratulations on this interesting work.

DISTRIBUTION OF MATERIALS:

Again, congratulations on a very nice paper. I hope you found the review process to be constructive and are pleased with how the manuscript was handled editorially. We look forward to future exciting submissions from your lab.

Sincerely,
